# Effect of Dietary Level of Beet Pulp, with or without Molasses, on Health Status, Growth Performance, and Carcass and Digestive Tract Traits of Rabbits

**DOI:** 10.3390/ani12233441

**Published:** 2022-12-06

**Authors:** Orlando Arce, Gilbert Alagón, Luis Ródenas, Eugenio Martínez-Paredes, Vicente Javier Moya, Concha Cervera, Juan José Pascual

**Affiliations:** 1Departamento de Ciencia Animal, Facultad de Ciencias Agrarias y Naturales, Universidad Técnica de Oruro, Av. 6 de Octubre 5715, Cas. Postal 49, Oruro 0401, Bolivia; 2Facultad de Agronomía y Zootecnia, Universidad Nacional de San Antonio Abad del Cusco, Ap. Postal 921, Cusco 08000, Peru; 3Institute for Animal Science and Technology, Universitat Politècnica de València, Camino de Vera 14, 46071 Valencia, Spain

**Keywords:** mortality, daily gain, dressing out, cecal activity

## Abstract

**Simple Summary:**

Beet pulp is a raw material widely used in feed for growing rabbits, because it is a good source of soluble fiber, and its inclusion is frequently associated with a reduction in the incidence of digestive disorders. Beet pulp may or may not be accompanied by the molasses that is also obtained in the beet sugar extraction process, but no information is available on the effect of molasses presence on the response of the rabbits. This work evaluates the effect of the inclusion of beet pulp, with or without molasses, on the growth performance, carcass, digestive tract, and fermentative profile of the cecum in growing rabbits. The results of the present work have shown that beet pulp linearly reduced the growth performance and carcass yield of growing rabbits and, although the inclusion of beet pulp could contribute to reducing the risk of digestive disorders, when beet pulp included molasses, even higher incidence was observed.

**Abstract:**

To evaluate the effect of dietary level of beet pulp, with or without molasses, on growth performance, a total of 470 28-day-old rabbits were used (614 ± 6 g). Animals were randomly allocated into five dietary treatment groups: Control, without beet pulp; BP_20_, and BP_40_ with 20 and 40% of beet pulp without molasses, respectively; and BPM_20_ and BPM_40_, with 20 and 40% of beet pulp with molasses, respectively. Daily feed intake (DFI) and average daily gain (ADG) were controlled at 28, 49, and 59 days of age. Carcass and digestive tract traits were also determined at 59 days of age. Mortality and morbidity were controlled daily. Mortality during the growing period was higher in BPM than in BP groups (+9.2%; *p* < 0.05). The higher the inclusion of beet pulp, the lower the DFI and ADG of animals (5.5 and 4.6% for every 20% inclusion, respectively; *p* < 0.001), as well as the dressing out percentage, the liver proportion, and the dissectible fat percentage of their carcasses. However, the best feed efficiency during the last 10 days was obtained with the BPM_40_ group. The higher the inclusion of beet pulp, the higher the weight of the empty gastrointestinal tract and cecum (+2.4 and +3.0 percentage points for every 20% inclusion, respectively; *p* < 0.001). In fact, a higher inclusion of beet pulp decreased the pH and dry matter and decreased the total volatile fatty acids content of cecum richer in acetic acid but poorer in propionic, isobutiric, isovaleric, and valeric acids. Stomach weight was lower, and the capric acid content in the cecum was higher in the BPM than in the BP group. The inclusion of beet pulp in the feed reduced the growth performance and carcass yield of growing rabbits, and an even higher incidence of digestive disorders was observed when beet pulp included molasses.

## 1. Introduction

Beet pulp is a by-product of the sugar industry from beet roots, which is commercialized dehydrated and granulated. These pellets may or may not include other by-products, mainly molasses and vinasse, which can alter their characteristics and nutritional value for animals [1]. The incorporation of beet pulp into rabbit diets has increased considerably since the appearance of Epizootic Rabbit Enteropathy (ERE). The incidence of this disease, of still unknown etiology, can be partially modulated by the chemical composition of the diet. It is well known that a reduction in protein and an increase in the fiber content of the diet at the expense of starch usually reduces the risk of digestive disorders associated with ERE [2,3,4]. From this fiber, the inclusion of the so-called soluble fiber reduces mortality and improves the intestinal mucosa [5,6].

Beet pulp is rich in soluble fiber, which can be partially digested and fermented in the digestive tract of rabbits, so its incorporation into feed induces a modulation of the cecal fermentative pattern and microbiota [7,8]. In this way, many studies associate its inclusion with greater digestive safety, since mortality and morbidity during the growing period decrease [9].

However, some studies also indicate that a high inclusion of beet pulp could reduce growth performance and carcass yield by increasing the cecum size of the growing rabbits [10,11]. Moreover, beet pulp is a raw material whose composition varies greatly from one source to another. One of the main sources of variation is the inclusion or not of the molasses (and other by-products in a minor proportion, as stillage) obtained by the crystallization of the sugars in the marketed pellets. Molasses is frequently used in the manufacturing of feed as a palatable (rich in sugars) and binding agent, but its inclusion in the beet pulp pellet increases electrolyte and non-assimilable carbohydrate content [12]. Therefore, it is important to continue conducting studies to better understand the variability of this raw material and its effect on the rabbits’ performance and health. 

Therefore, the main aim of this study was to evaluate the effect of dietary beet pulp inclusion level, with and without molasses, on the health status, growth performance, carcass, digestive tract, and fermentative profile of the cecum in growing rabbits under a ERE outbreak when fed with medicated feeds.

## 2. Materials and Methods

### 2.1. Diets

Two batches of beet pulp, with (BPM) and without molasses (BP) obtained during sugar manufacture, were utilized, from the same batch of beet pulp from an industrial sugar factory. Starting from a control diet (C, without beet pulp) formulated to fit the requirements of growing rabbits [1], another four experimental diets were designed, including 20 (BP_20_) and 40% (BP_40_) of BP and 20 (BPM_20_) and 40% (BPM_40_) of BPM (Table 1). Diets were formulated to be iso-protein, iso-fibrous, and iso-energetic, using the average chemical composition and nutritive value of the beet pulps analyzed in [13]. 

The chemical composition and nutritive value of the diets are also presented in Table 1. Experimental diets were iso-protein (crude protein (CP), ranging from 170 to 173 g/kg dry matter (DM)), and iso-fibrous (neutral detergent fiber from 395 to 415 g NDF/kg DM). However, digestible protein (DP) and digestible energy (DE) ranged from 120 to 143 g/kg DM and from 11.7 to 12.5 MJ/kg DM, respectively. For every 10% inclusion of BP, the neutral detergent soluble fiber (NDSF) content increased by 13 g/kg DM. As the experiment was carried out during an ERE outbreak, all experimental feeds (up to 49 days of age) included a mixture of antimicrobials prescribed by the vet to avoid an excessive incidence of ERE (29 ppm dincomicina, 29 ppm spectinomicyn, 120 ppm neomycin, and 50 ppm tiamulin). Although most commercial farms require the use of antimicrobials to control the incidence of ERE, they can modulate the microbiota and affect the response of the animals, so the results of this work should be framed in this context. In any case, all of the treatments received the same prescription, so the results are comparable and useful in the current context.

### 2.2. Animals

The experimental protocols followed both the Spanish Royal Decree 53/2013 on the protection of animals used for scientific purposes [14] and the recommendations for applied nutrition research in rabbits, described by the European Group on Rabbit Nutrition [15] and approved by the Committee of Ethics and Animal Welfare of the Universitat Politècnica de València.

A total of 470 three-way crossbred rabbits were used. The animals were weaned at 28 days of age, with an average body weight (BW) of 612.2 ± 103.4 g, and were blocked by litter. Weaned rabbits from each litter were randomly assigned to the different treatments (94 animals per treatment) in each batch (a total of 5 batches from January to July 2014). All the animals were individually housed in boxes (26 × 50 × 31 cm) until the end of the growing period (59 days of age). The rabbit was considered an experimental unit, and both male and female animals were used. To determine DE and DP of the diets, another group of 50 animals (10 per diet; first batch) were housed at 42 days of age in metabolic cages. Feed intake and total feces production were controlled and sampled from 49 to 53 days of age [16]. Feces samples were proportionally pooled to obtain a sample per diet for further analysis. Feeds and feces were stored and analyzed to determine DM, CP, and gross energy (GE).

### 2.3. Experimental Procedure

BW and feed intake were monitored on days 28, 49, and 59 of age to determine daily feed intake (DFI), average daily gain (ADG), and feed conversion ratio (FCR). Mortality and morbidity were controlled daily, and sanitary risk index was calculated as the sum of mortality and morbidity [17]. Undead animals were classified as morbid when showing diarrhea, constipation, immobility, and weight loss, or decreased feed intake compared with animals receiving the same treatment (under 1.5 interquartil ranges; SAS, 2002). 

At 59 days of age, 100 rabbits (20 per diet) were weighed (slaughter weight, SW), electrically stunned, and slaughtered at the abattoir on the farm. No fasting was applied. The slaughtering and carcass dissection procedures followed the recommendations of [18]. The slaughtered rabbits were bled, and the skin, genitals, urinary bladder, full gastrointestinal tract, and distal part of the legs were removed. After measuring the pH of both stomach and cecal contents, aliquots of approximately 1 g of cecal content were weighed, and 3 mL of a solution of 2% sulfuric acid or 2 mL of 2% ortho-phosphoric acid was added for further analysis of ammonia nitrogen (N-NH3) and volatile fatty acids (VFA), respectively. Samples for VFA analysis were centrifuged at 10,000 rpm for 10 min, and the liquid phase was collected into Eppendorf vials of 0.5 mL. Finally, all samples were stored at −80 °C until analysis. The remaining cecal content was stored at −20 °C until DM analysis. The empty gastrointestinal tract (EGT) was weighed and expressed as a percentage compared to SW. Subsequently, both empty stomach and cecum were weighed and expressed as percentages compared to EGT. The hot carcasses obtained were weighed (HCW) and then chilled at +4 °C for 24 h in a ventilated room. The chilled carcasses were weighed (CCW), and the dressing out percentage (DoP) was calculated as CCW × 100/SW. Liver and dissectible fat (inguinal fat, perirenal fat, and scapular fat) were removed, weighed, and expressed as percentages compared to CCW (LvP and DFaP, respectively). 

### 2.4. Chemical Analysis

Chemical analyses of diets and feces were performed following the AOAC methods [19]: 934.01 for DM, 942.05 for ash, 976.06 for crude protein (CP), and 920.39 with previous acid hydrolysis of samples for ether extract (EE). Starch content was determined by a 2-step enzymatic procedure, with solubilization and hydrolysis to maltodextrins with thermostable α-amylase, followed by complete hydrolysis with amyloglucosidase (both enzymes from Sigma-Aldrich, Steinheim, Germany). The resulting glucose was measured by the hexokinase/glucose-6 phosphate dehydrogenase/NADP system (R-Biopharm, Darmstadt, Germany). Neutral detergent fiber (NDF), ADF, and acid detergent lignin (ADL) fractions were analyzed sequentially [20], with a thermo-stable α-amylase pre-treatment and expressed exclusive of residual ash, using a nylon filter bag system (Ankom, Macedon, NY, USA). Neutral detergent soluble fiber (NDSF) content was determined according to [21] and modified by [4]. CP content of NDF residue was analyzed to determine the protein associated with the NDF (CP-NDF). Gross energy was determined by combustion in adiabatic calorimetric pump, according to the recommendations of the European Group on Rabbit Nutrition (EGRAN) [22]. The content of the main limiting amino acids (lysine, methionine, and threonine) was determined after acid hydrolysis with HCL 6N at 110 °C for 23 h by HPLC, as described in [23]. Gross energy of the diets and the pooled feces was determined using a calorimetric bomb.

The DM and N-NH3 concentrations in the cecal contents were determined according to AOAC [19] procedures (methods 934.01 and 984.13, respectively). For VFA analysis, samples were previously filtered through a cellulose filter (0.45) and 250 mL was transferred to the injection vials. Two microliters from each sample were injected into the gas chromatograph (FISONS 8000 series, Milan, Italy), equipped with an AS800 automatic injector. The column used was a BD-FFAP of 30 m length × 0.25 mm internal diameter × 0.25 mm film thickness. The injector and detector temperatures were maintained at 220 and 225 °C, respectively.

### 2.5. Statistical Analysis

Mortality, morbidity, and sanitary risk index during the growing period were analyzed using a logistic regression, by the GENMOD procedure of the Statistical Analysis System (SAS, [24]), considering a binomial distribution. The results were transformed from the logit scale. 

Data on performance traits were analyzed using a GLM procedure from SAS [24]. The model included, as fixed effects, the diet (C, BP_20_, BP_40_, BPM_20_, BPM_40_), the batch (1, 2, 3, 4, 5), and their interaction. The litter was included as a block, and BW at 28 d of age as a covariate. Data on digestive tract and cecal traits were also analyzed using a GLM procedure for each age, with a model including the diet, the batch, and their interaction as the main effect. The effect of molasses inclusion was tested by orthogonal contrasts [½ (BP_20_ + BP_40_) – ½(BPM_20_ + BPM_40_)]. Linear and quadratic effects for the beet pulp inclusion (0, 20, and 40%) were analyzed by polynomial contrasts. All data were presented as least-squares means, and for mean comparison, a t-test was used. Although the batch had an effect in most analyses, no interaction of the diet with the batch was observed.

## 3. Results

### 3.1. Health Status

As can be seen in Table 2, where the main controlled health parameters are presented, the trial took place during an ERE outbreak in the experimental farm, with high mean values of mortality, morbidity, and SRI (on average, 25.5, 5.5, and 31.0%, respectively). The inclusion of beet pulp in the feed had no significant effect on the mean values of mortality and morbidity of growing rabbits. However, mortality observed with BPM feeds was significantly higher than with BP feeds, both from 49 to 59 d of age (−10.8%; *p* = 0.0207) and throughout the growth period (−9.2%; *p* = 0.0321). 

Figure 1 shows us the weekly evolution of the mortality of the growing rabbits with the various feeds throughout the fattening period. As can be observed, there were no differences between the treatments up to 49 d of age, but from that moment, the feeds that included a higher level of BPM begin to show a higher incidence of mortality than those that included a higher level of BP. In fact, the group fed BPM40 showed significantly higher mortality than those fed BP40 at the end of the trial (+14.2%; *p* < 0.0012).

### 3.2. Growth Performance

Table 3 shows the effect of beet pulp inclusion, with or without molasses, on the growth performance of the rabbits during the fattening period (28 to 59 d of age). Regardless of the inclusion or not of molasses, for every 20% inclusion of beet pulp in the feed, there was a linear reduction in the DFI and ADG of 5.5 and 4.6%, respectively, throughout the entire fattening period (*p* < 0.0001). From 28 to 49 d of age, lower ADG and BW at 49 d of age were observed in growing rabbits with beet pulp inclusion (−6.0% and −4.2% for each 20% inclusion of BP or BPM in the diet, respectively; *p* < 0.0001), without the FCR being affected in that period. However, during the last 10 days of the growing period, no significant effect of feed on ADG was observed, which allowed a reduction in FCR from 49 to 59 d of age with beet pulp inclusion, although this was only observed and significant with the molasses inclusion (−6.4% for each 20% inclusion of BPM in the diet; *p* = 0.0135). 

### 3.3. Carcass and Digestive Tract

The possible effect of the inclusion of BP, with or without molasses, on the main carcass traits of rabbits at 59 d of age is shown in Table 4. No significant differences were observed in SW at 59 d of age. However, HCW and CCW linearly decreased by −3.9% for each 20% inclusion of BP or BMP in the diet (*p* = 0.0007). Regardless of the inclusion or not of molasses, DoP, LvP, and DFaP decreased linearly with the inclusion of beet pulp (−1.8, −0.72, and −0.87 points of percentage for every 20% inclusion of BP or BPM in the diet, respectively; *p* < 0.001). On the contrary, the inclusion of beet pulp significantly increased the percentages of EGT and cecum of the animals (+ 2.4 and +3.0 percentage points for every 20% inclusion of BP or BPM in the diet, respectively; *p* < 0.0001). The stomach percentage was significantly higher for BP than BPM animals (+1.9 percentage points; *p* = 0.0120).

Regarding the functionality of the digestive tract at 59 d of age (Table 5), the inclusion of beet pulp did not affect the pH of the stomach, but led to a linear reduction in the pH of the cecum (−0.10 points for every 20% inclusion of BP or BPM in the diet; *p* = 0.0055), and to both linear and quadratic (especially with the first 20% inclusion) decreases in the cecum DM percentage (−2.5 percentage points for every 20% inclusion of BP or BPM in the diet; *p* = 0.0001). NH_3_-N production in the cecum was not affected by the feed, but total VFA production increased linearly and quadratically with the inclusion of beet pulp (+17% with the first 20% inclusion of BP or BPM in the diet; *p* = 0.0131). Regarding the proportion of the different VFAs of the cecum, the inclusion of beet pulp in the feed linearly increased the percentage of acetic acid (*p* = 0.0024), linearly reduced the percentage of propionic acid (*p* = 0.0289), and linearly and quadratically reduced (mainly with the first 20% of inclusion) the percentage of isobutyric (*p* = 0.0019) and isovaleric acid (*p* = 0.0002). Finally, the caproic acid percentage of cecum was reduced with BP inclusion (*p* = 0.0198), but it was not affected by the BPM inclusion.

## 4. Discussion

Sugar beet pulp has become one of the most common raw materials in the formulation of rabbit feeds in recent decades. On average, beet pulps are usually included at levels close to 100 g/kg in commercial feeds [1], but their level in peri-weaning feeds is even higher (on av. 160 g/kg, ranging in experimental diets from 0 to 490 g/kg; [6]). However, one of the concerns of the sector when using beet pulp is the possible heterogeneity of the quality of this by-product. As mentioned above, its main source of variability is the inclusion or not in the marketed pellet of the molasses (and other by-products in a minor proportion included in this molasses, as stillage). The present work has faced, for the first time, how the inclusion of high levels of beet pulp, as well as the presence or absence of molasses obtained in its manufacture, can affect health status, growth performance, carcass, digestive tract, and fermentative profile of the cecum in growing rabbits.

### 4.1. Health Status

The trial took place during an ERE outbreak in the experimental farm, which led the veterinary service to prescribe an antibiotic treatment in all experimental feeds from weaning to 49 days of age. During the application of antibiotic treatment, most groups showed mortality close to 9%, except the BPM_40_ group, for which it was above 13%. However, once the treatment was withdrawn, mortality increased in C and BPM groups (close to 18%), while the inclusion of BP linearly reduced mortality compared to the control. Therefore, an interaction between the inclusion level of beet pulp and the inclusion of molasses in the mortality of growing rabbits in the global period was observed. 

Kpodékon et al. [25], with a reduced number of animals per treatment, did not observe any effect of the dietary inclusion of 5% molasses on the digestive health of growing rabbits. However, many studies have observed a reduction in mortality associated with ERE when more fermentable soluble and insoluble fiber (usually from sugar beet pulp) was introduced, at the expense of reducing the starch or other fiber fractions of the diet [4,7,26,27,28,29,30,31,32,33,34,35]. In fact, as described in the review by [6], the incidence of mortality is reduced by 2.5% for every 1% increase in soluble fiber in the diet. However, in other works carried out in experimental [36,37,38] or field conditions [29], the increase in the soluble fiber of the feed by including beet pulp did not have a relevant effect on the health status of fattening rabbits. This lack of response may be due to different factors, such as the number of animals per treatment, inclusion level of beet pulp, good health status of the experimental farm, etc. In any case, based on the results of this work, in this list of possible factors, we should also include the quality of the beet pulp used. When including beet pulp with the aim of improving the sanitary state, it would be advisable to search for a source of beet pulp that is not pelleted with the inclusion of molasses and stillage, which are also produced in sugar manufacturing.

The differences in digestive health observed with the different diets may be related to the digestive environment generated in the large intestine of growing rabbits. Regardless of the inclusion or not of molasses, the inclusion of beet pulp in the diets linearly reduced the pH of the cecum, due to an increase in the production of VFA, showing a quadratic reduction in N-NH3. These results have been previously observed by other authors [11,28,30,31,35], who maintain that the greater availability of fermentable substrates, extensively used by the microbiota of the cecum, would intensify and stabilize the microbial activity in the cecum of rabbits. This greater availability of fermentable substrate promotes the development of the fibrolytic versus the amylolytic microbiota, which increases the proportion of acetic acid at the expense of the rest of the short-chain organic acids [39]. This possible improvement in the microbiota, together with other factors associated with an improvement in the digestive environment, such as the increase in the villi height to crypt depth ratio [7] and the production of mucins from a greater number of Globet cells per villi [40], could be the reason for the usual improvement in the digestive health status of growing rabbits when beet pulp is added to the feed.

However, contrary to that expected, when the beet pulp also included the molasses and the stillage obtained in its manufacturing, an increase in digestive incidences was observed. In general, no significant differences were observed in the fermentative profile depending on the inclusion or not of molasses in the beet pulp used. However, diets rich in BPM had higher CP associated to the NDF, and reduced protein digestibility. This may increase ileal N flow to the cecum and proliferation of some *Clostridium* species, which have been associated with increased mortality in growing rabbits [41]. In fact, the only difference observed in the fermentation pattern was the linear increase in caproic acid content in the cecum of animals with the BPM inclusion, while this decreased linearly with the inclusion of BP. Caproic acid formation occurs through a carboxylic acid chain elongation process, which uses reverse β-oxidation of acetic and/or n-butyric acid, and ethanol or lactic acid as an electron donor. *Clostridium* spp. has been identified as a determinant when ethanol was provided as electron donor [42]. Therefore, the production of caproic acid in the cecum could be an indicator of the greater proliferation of *Clostridium* spp.

### 4.2. Growth Performance

After weaning, the inclusion of beet pulp in the diet, both with and without molasses, led to a linear reduction in the DFI of growing rabbits during the entire fattening period. Most of the previous works have also observed that, when the inclusion level of beet pulp is increased, usually at the expense of the dietary starch content, a reduction in the DFI of growing rabbits is observed [4,11,30,34,37,39,43]. However, other studies have not found significant differences in DFI with the addition of beet pulp [26,27,32,33], and even an increase in consumption associated to an increase in the level of ADF [31]. Sugar beet pulp is especially rich in pectin, pentoses, and beta-glucans. Gidenne et al. [44] described that these soluble fiber constituents have high water-holding capacity and gel formation, which would increase the weight of gastric and cecal content, slowing down digestive transit and, consequently, reducing the ingestion capacity of growing rabbits. Proof of the water-holding capacity of beet pulp in the digestive tract of rabbits was the clear linear increase in the water content of the cecum in the rabbits in this study (from 75 to 80%). On the other hand, this effect was also clearly manifested by the linear increase in the proportion of EGT with the inclusion of beet pulp (from 18 to even 23% of the LW), which is mainly due to a linear increase in the proportion of cecum of the EGT.

Although the diets were formulated to be isoenergetic, using the nutritive values obtained in previous work [13], DE and DP levels of the beet pulp batches used in this work seemed to be lower than expected, especially with those that included molasses (Table 1). Because of the lower DFI and the slightly lower nutritional value, the dietary inclusion of beet pulp led to a linear reduction in the ADG of growing rabbits during the first two weeks post-weaning. These results agree with some previous works [11,29,35]. However, most studies (regardless of DFI observed) did not observe significant effects on ADG with the addition of beet pulp [4,26,27,30,31,32,33,34,35,36,38]. This could be related to the increased retention of digestive content in animals fed feed rich in beet pulp. Therefore, although no significant differences were observed in ADG, the value obtained with diets rich in beet pulp would not be solely due to the growth of the animals. In fact, as observed in this study, several studies showed a reduction in the DoP and weight of the carcasses of rabbits fed diets rich in beet pulp, since part of their ADG was addressed to increase the digestive content of these animals.

In this way, the possible improvements that can be seen in feed efficiency in this work and in previous ones associated with diets rich in beet pulp [11,30,34,35,36,38] would be fictitious, due to the effect that this type of pulp has on ingestion, development of the digestive tract, and carcass yield. In addition, the DE from soluble fiber has a lower metabolic rate than that from soluble carbohydrates, which could explain why animals fed diets rich in beet pulp had a lower percentage of liver and dissectible fat in their carcasses. These results agree with those obtained by [35,38]. Therefore, although the performance traits obtained with diets rich in beet pulp may seem similar to those obtained with diets richer in starch sources, the final consequences on the yield obtained in their carcasses should make us consider their use limitation, or at least take it into account when determining the final commercial weight of the desired carcasses.

## 5. Conclusions

Sugar beet pulp is a good source of soluble fiber, which, when used properly, can help to improve the digestive health of growing rabbits after weaning. However, the results of this study have shown that it is very important to take into account the quality and inclusion level of the beet pulp used to obtain the desired results. The use of a beet pulp manufactured with its molasses (and other by-products in a minor proportion, as stillage) is characterized by having a lower digestibility of its protein, and even increasing the incidence of digestive disorders in an ERE outbreak. On the other hand, due to its ability to increase the total content of the digestive tract and the lower metabolic rate of its soluble fiber, the inclusion level of beet pulp should be limited in order to avoid excessive deterioration of the carcass yield.

## Figures and Tables

**Figure 1 animals-12-03441-f001:**
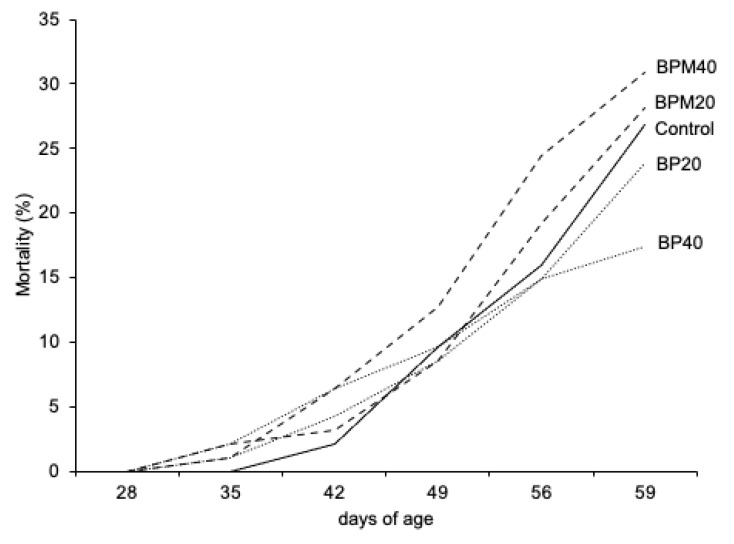
Evolution of mortality of growing rabbits during the fattening period as a function of the experimental diet. C, control without beet pulp; BP_20_, with 20% of beet pulp without molasses; BP_40_, with 40% of beet pulp without molasses; BPM_20_, with 20% of beet pulp with molasses; BPM_40_, with 40% of beet pulp with molasses.

**Table 1 animals-12-03441-t001:** Ingredients (%) and chemical composition (g/kg DM) of the experimental diets ^1^.

	C	BP_20_	BP_40_	BPM_20_	BPM_40_
Ingredients					
Barley grain	14.00	7.00	-	7.00	-
Beet pulp without molasses	-	20.00	40.00	-	-
Beet pulp with molasses	-	-	-	20.00	40.00
Wheat bran	13.00	13.00	13.00	12.00	11.00
Beet pulp molasses	2.00	1.00	-	1.75	1.50
Sunflower meal (30% CP)	12.00	13.00	14.00	14.00	16.00
Soybean meal (44% CP)	11.00	10.00	9.00	10.50	10.00
Alfalfa hay	16.00	16.00	16.00	13.50	11.00
Grape seed	9.00	7.00	5.00	7.00	5.00
Oat hulls	10.00	5.00	-	5.00	-
Soybean oil	5.00	2.75	0.50	4.10	3.20
Starch	5.00	2.50	-	2.50	-
Dicalcium carbonate	0.50	0.25	-	0.25	-
Dicalcium phosphate 2·H_2_O	0.80	0.85	0.90	0.83	0.86
Sodium chloride	0.40	0.40	0.40	0.40	0.40
DL-methionine	0.06	0.10	0.13	0.03	-
L-lysine	0.20	0.16	0.13	0.15	0.10
L-threonine	0.14	0.09	0.04	0.09	0.04
Vitamin-mineral premix ^2^	0.50	0.50	0.50	0.50	0.50
Robenidine HCl ^3^	0.10	0.10	0.10	0.10	0.10
Antibiotic premix ^4^	0.30	0.30	0.30	0.30	0.30
Chemical composition					
Dry matter, DM	914	914	914	915	912
Ash	60	63	65	63	65
Ether extract	66	45	26	56	48
Starch	152	105	55	102	40
Neutral detergent fiber, NDF	400	395	394	406	415
Acid detergent fiber	213	227	240	234	251
Acid detergent lignin	65	59	52	59	52
Neutral detergent soluble fiber	170	199	220	204	227
Crude protein, CP	172	173	173	170	173
CP-NDF	43	44	35	44	52
Digestible protein, DP	143	140	133	135	120
Gross energy (MJ/kg DM)	20.1	19.2	18.7	19.0	18.6
Digestible energy, DE (MJ/kg DM)	12.5	12.2	11.9	11.8	11.7
DP/DE (g DP/MJ DE)	11.5	11.5	11.2	11.4	10.3

^1^ Experimental diets: C, control without beet pulp; BP_20_, with 20% of beet pulp without molasses; BP_40_, with 40% of beet pulp without molasses; BPM_20_, with 20% of beet pulp with molasses; BPM_40_, with 40% of beet pulp with molasses. ^2^ Vitamin-micromineral mixture supplies per kg of feed: Vitamin A: 8.375 IU; Vitamin D3: 750 IU; Vitamin E: 20 mg; Vitamin K3: 1 mg; Vitamin B1: 1 mg; Vitamin B2: 2 mg; Vitamin B6: 1 mg; Nicotinic acid: 20 mg; Choline chloride: 250 mg; Magnesium: 290 mg; Manganese: 20 mg; Zinc: 60 mg; Iodine: 1.25 mg; Iron: 26 mg; Copper: 10 mg; Cobalt: 0.7 mg; Butyl hydroxylanysole and ethoxiquin mixture: 4 mg. ^3^ Robenidine at 66 g/kg. ^4^ Dinco-spectim (29ppm dincomicina + 29ppm spectinomicyn), 120ppm neomicin, Apsamix Tiamulin (50ppm tiamulin), normally used in rabbit farms with a high incidence of mucoid enteropathy. CP: crude protein.

**Table 2 animals-12-03441-t002:** Effect of beet pulp inclusion (with or without molasses) on the mortality and morbidity of fattening rabbits.

	Experimental Diets ^1^					*p*-Value			
	C	BP_20_	BP_40_	BPM_20_	BPM_40_	BP-BPM	Feed	Linear	Quadratic
No. animals	94	94	94	94	94				
Mortality (%):									
28–49 d	9.75	9.95	8.87	8.86	13.22	0.6034	0.8383	0.7234	0.7228
49–59 d	17.1	13.9	8.6	19.3	18.4	0.0207	0.2454	0.4254	0.7131
28–59 d	26.9	23.8	17.4	28.1	31.6	0.0321	0.2203	0.6669	0.9447
Morbidity (%):									
28–59 d	3.99	5.04	8.25	8.20	1.87	0.5185	0.2531	0.7150	0.3480
SRI (%):									
28–59 d	30.9	28.9	25.7	36.3	33.5	0.1180	0.5682	0.8272	0.5948

^1^ Experimental diets: C, control without beet pulp; BP_20_, with 20% of beet pulp without molasses; BP_40_, with 40% of beet pulp without molasses; BPM_20_, with 20% of beet pulp with molasses; BPM_40_, with 40% of beet pulp with molasses. Contrast BP-BPM: (BP_20_ + BP_40_)/2 vs. (BPM_20_ + BPM_40_)/2. SRI: sanitary risk index (mortality + morbidity).

**Table 3 animals-12-03441-t003:** Effect of beet pulp inclusion (with or without molasses) on the growth performance of fattening rabbits.

	Experimental Diets ^1^						Contrast		*p*-Value		
	C	BP_20_	BP_40_	BPM_20_	BPM_40_	SEM	BP-BPM		Feed	Linear	Quadratic
No. of animals	65	67	73	57	63						
Body weight (g):											
28 d	618	604	609	626	603	12	–8	±11	0.5377	0.3500	0.7524
49 d	1741 ^c^	1678 ^b^	1621 ^a^	1696 ^bc^	1586 ^a^	19	8	±18	0.0001	0.0001	0.3802
59 d	2202 ^c^	2135 ^b^	2063 ^a^	2148 ^bc^	2045 ^a^	24	3	±22	0.0001	0.0001	0.5227
Daily feed intake (g DM/d):											
28–49 d	106 ^d^	99 ^bc^	95 ^ab^	100 ^c^	91^a^	2	1	±1	0.0001	0.0001	0.9937
49–59 d	146 ^c^	139 ^b^	136 ^b^	137 ^b^	128 ^a^	3	5	±3	0.0001	0.0001	0.7455
28–59 d	119 ^c^	112 ^b^	108 ^b^	112 ^b^	103 ^a^	2	2	±2	0.0001	0.0001	0.8701
Average daily gain (g/d):											
28–49 d	54.1 ^c^	51.1 ^b^	48.3 ^a^	51.9 ^bc^	46.7 ^a^	0.9	0.4	±0.9	0.0001	0.0001	0.3830
49–59 d	46.1	45.7	44.2	45.2	45.9	1.2	–0.6	±1.2	0.7482	0.4459	0.8847
28–59 d	51.5 ^c^	49.3 ^b^	47.0 ^a^	49.7 ^bc^	46.4 ^a^	0.8	0.1	±0.7	0.0001	0.001	0.5222
Feed conversion ratio ^2^:											
28–49 d	2.22	2.16	2.23	2.17	2.20	0.03	0.01	±0.03	0.3935	0.7873	0.0649
49–59 d	3.59 ^b^	3.46 ^b^	3.54 ^b^	3.52 ^b^	3.13 ^a^	0.07	0.18	±0.07 *	0.0001	0.0050	0.7196
28–59 d	2.31	2.27	2.29	2.26	2.21	0.03	0.05	±0.03	0.0718	0.0642	0.6062

^1^ Experimental diets: C, control without beet pulp; BP_20_, with 20% of beet pulp without molasses; BP_40_, with 40% of beet pulp without molasses; BPM_20_, with 20% of beet pulp with molasses; BPM_40_, with 40% of beet pulp with molasses. Contrast BP-BPM: [(BP_20_ + BP_40_)/2 – (BPM_20_ + BPM_40_)/2]; ^2^ feed conversion ratio (g/g) was calculated as the ratio of daily feed intake (in fresh basis) with respect to the average daily gain. * *p* < 0.05. ^a–d^: means not sharing superscript within a row were significantly different at *p* < 0.05. SEM: standard error of the mean.

**Table 4 animals-12-03441-t004:** Effect of beet pulp inclusion (with or without molasses) on the carcass traits of fattening rabbits at 59 d of age.

	Experimental Diets ^1^						Contrast		*p*-Value		
	C	BP_20_	BP_40_	BPM_20_	BPM_40_	SEM	BP-BPM		Feed	Linear	Quadratic
No. of animals	40	40	40	40	40						
Slaughter weight (SW) (g)	2142	2108	2083	2099	2100	26	3.6	±26	0.6106	0.1232	0.6074
Hot carcass weight (g)	1281 ^b^	1198 ^a^	1199 ^a^	1183 ^a^	1164 ^a^	24	25	±23	0.0076	0.0007	0.0565
Cold carcass weight (CCW) (g)	1228 ^b^	1151 ^a^	1150 ^a^	1129 ^a^	1111 ^a^	24	30	±23	0.0063	0.0007	0.0612
Dressing out percentage (%)	59.7 ^c^	57.9 ^b^	56.5 ^a^	57.3 ^ab^	55.9 ^a^	0.5	0.6	±0.5	0.0001	0.0001	0.4513
Liver weight (% CCW)	6.75 ^c^	6.37 ^bc^	5.30 ^a^	5.59 ^ab^	5.34 ^a^	0.36	0.36	±0.35	0.0168	0.0016	0.8595
Dissectible fat (% CCW)	4.87 ^c^	3.77 ^b^	2.83^a^	3.80 ^b^	3.45 ^ab^	0.23	0.05	±0.03	0.0001	0.0001	0.2882
EGT (%LW)^2^	17.6 ^a^	20.2 ^b^	21.9 ^cd^	20.8 ^bc^	22.9 ^d^	0.5	0.9	±0.5	0.0001	0.0001	0.3143

^1^ Experimental diets: C, control without beet pulp; BP_20_, with 20% of beet pulp without molasses; BP_40_, with 40% of beet pulp without molasses; BPM_20_, with 20% of beet pulp with molasses; BPM_40_, with 40% of beet pulp with molasses. Contrast BP-BPM: [(BP_20_ + BP_40_)/2 − (BPM_20_ + BPM_40_)/2]. Dissectable fat: scapular + inguinal + perirenal fat. 2EGT: empty gastrointestinal tract. ^a–d^: means not sharing superscripts within a row were significantly different at *p* < 0.05. SEM: standard error of the mean.

**Table 5 animals-12-03441-t005:** Effect of beet pulp inclusion (with or without molasses) on digestive tract traits of fattening rabbits at 59 d of age.

	Experimental Diets ^1^						Contrast		*p*-Value		
	C	BP_20_	BP_40_	BPM_20_	BPM_40_	SEM	BP-BPM		Feed	Linear	Quadratic
No. of animals	40	40	40	40	40						
Stomach traits:											
Weight (% EGT)	23.0	24.9	24.4	22.8	22.7	0.7	1.9	±0.7 *	0.1185	0.5392	0.3848
pH	1.41	1.44	1.41	1.38	1.34	0.04	0.06	±0.04	0.3302	0.3220	0.5979
Cecum traits:											
Weight (% EGT)	32.2 ^a^	35.5 ^b^	39.1 ^c^	36.1 ^b^	37.1 ^bc^	1.1	–0.7	±1.1	0.0003	0.0001	0.4968
pH	6.18 ^b^	6.06 ^ab^	5.96 ^a^	6.03 ^ab^	5.99 ^a^	0.06	0.00	±0.06	0.0909	0.0055	0.5810
DM (%)	25.5 ^c^	21.4 ^b^	19.9 ^a^	21.4 ^b^	20.2 ^ab^	0.4	0.1	±0.4	0.0001	0.0001	0.0011
N-NH_3_ (mmol/L)	9.32	5.58	7.42	6.70	7.27	1.25	0.48	±1.22	0.2990	0.2023	0.0574
Total VFA (mmol/L)	73.1 ^a^	85.3 ^b^	87.2 ^b^	85.8 ^b^	85.6 ^b^	4.1	–0.6	±4.2	0.1030	0.0101	0.0131
Acetic acid (%)	76.0 ^a^	79.6 ^b^	80.3 ^b^	78.6 ^b^	78.5 ^b^	0.9	–1.3	±0.9	0.0143	0.0024	0.1024
Propionic acid (%)	4.98 ^c^	4.48 ^ab^	4.14 ^a^	5.42 ^b^	3.98 ^a^	0.32	0.39	±0.34	0.0209	0.0289	0.1745
Butyric acid (%)	16.8	14.0	13.8	14.0	15.8	1.0	1.0	±1.0	0.1302	0.1055	0.0565
Isobutyric acid (%)	0.23 ^b^	0.10 ^a^	0.14 ^a^	0.10 ^a^	0.12 ^a^	0.03	–0.01	±0.03	0.0040	0.0024	0.0019
Isovaleric acid (%)	0.26 ^b^	0.10 ^a^	0.14 ^a^	0.10 ^a^	0.12 ^a^	0.03	–0.01	±0.03	0.0002	0.0001	0.0002
Valeric acid (%)	0.71 ^c^	0.46 ^ab^	0.40 ^a^	0.55 ^b^	0.42 ^a^	0.04	0.05	±0.04	0.0001	0.0001	0.1807
Caproic acid (%)	0.66 ^bc^	0.53 ^abc^	0.37 ^a^	0.65 ^bc^	0.73 ^c^	0.08	0.24	±0.07 *	0.0198	0.2784	0.8246
Heptanoic acid (%)	0.04	0.04	0.08	0.04	0.05	0.01	0.01	±0.02	0.2250	0.2706	0.2223

^1^ Experimental diets: C, control without beet pulp; BP_20_, with 20% of beet pulp without molasses; BP_40_, with 40% of beet pulp without molasses; BPM_20_, with 20% of beet pulp with molasses; BPM_40_, with 40% of beet pulp with molasses. Contrast BP-BPM: [(BP_20_ + BP_40_)/2 − (BPM_20_ + BPM_40_)/2]. * *p* < 0.05. ^a–c^: means not sharing superscript within a row were significantly different at *p* < 0.05. SEM: standard error of the mean.

## Data Availability

Not applicable.

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
