# Peer review of "Effect of Dietary Level of Beet Pulp, with or without Molasses, on Health Status, Growth Performance, and Carcass and Digestive Tract Traits of Rabbits"

_animals, 2022, doi:10.3390/ani12233441_

Round 1

Reviewer 1 Report (Previous Reviewer 3)

The authors did a good job addressing all the prior concerns. I have no further concerns.

Reviewer 2 Report (Previous Reviewer 2)

Dear authors,

I red your answer and I can see your point of view. Although I still believe that the doses could have been arranged better, I have decided to allow your manuscript to be published. Regarding the first sentence of the conclusion, after your explanation, I think you are right.

I wish you good luck in the further stages of publishing this manuscript! 

Reviewer 3 Report (Previous Reviewer 1)

We analyzed the responses given by the authors and the changes made to the paper. We are of the opinion that the work has improved a lot and is now ready to be accepted for publication.

This manuscript is a resubmission of an earlier submission. The following is a list of the peer review reports and author responses from that submission.

Round 1

Reviewer 1 Report

Consult the attached document

Reviewer 2 Report

In my opinion, much effort has gone into the experiment and the manuscript has been written quite carefully, but it does not deserve publication in the journal Animals.

The main concern that puts the experience into question is that the authors have quite heavily manipulated the levels of barley grain, sunflower meal, alfalfa hay, grape seed, oat hulls and soybean oil. If the authors manipulated diets with only one group (e.g. cereals or soybean/sunflower meal), the experience could be justified by substituting one with the other, but at present it is impossible.

- M&M diets: please explain terms BP and BPM in the text.

- Conclusions: First sentence is not a result from your study. You should delete it.

Reviewer 3 Report

Thank you for inviting me to review this paper by Arce et al.

Digestive disorders are the main factor responsible for reduced performance and health in growing rabbits, especially in recent decades when the incidence of Epizootic Rabbit Enteropathy (ERE) has increased. The inclusion of antimicrobials has often controlled the negative effects of ERE, but has increased the health costs. Moreover, frequent usage of antimicrobials increases the risk of presence of residuals in the meat and impairs consumer perception towards intensive rabbit farming. It is well known that adequate nutrition and feeding strategies can minimize the risk of these disorders. Among these strategies, the beneficial effect of increasing highly digestible fiber and reducing starch on the digestive health of growing rabbits is well established and consequently widely used. Highly digestible fiber promotes fermentative activity and induces beneficial changes in the caecal environment. In the present study authors evaluated  the effects of dietary beet pulp inclusion level, with and without molasses, on the health status, growth performance, carcass, digestive tract, and fermentative profile of the caecum in growing rabbits. This study is highly topical .

Short chain fatty acids are the main metabolite produced by intestinal bacteria during fiber fermentation and their abundance in the digestive tract reflects the level of intestinal fermentation. Fermentative substrates of some species serve as substrates for fermentation or are incorporated as intermediate metabolites into the metabolic pathways of other species which results in gradual fermentation of substrates. With regard to the differences in the mechanisms of ATB action, different ATB may selectively inhibit different members of the gut microbiota. Administration of various antimicrobials based on non-defined individual doses of the individual components per animal presents the risk of the absence of control of the exact effect of the antibiotic treatment. It is exactly the persisting microbiota of the animals treated with antibiotics that can have a decisive influence on colonization of the intestine and can result in increased counts of some species with considerable negative influence on the animal health, not considering the fact that this represents an uncontrolled situation with negative influence on reproducibility of studies (for example) concerning the effect of dietary on fermentative activity of digestive tract.

Although the authors of the present study mentioned this fact in the (Materials and Methods) section, they elaborate their conclusions without including the fact that the treatment of experimental animals with antimicrobial substances had a decisive negative effect and can influence the digestive and absorptive capacity of the digestive tract. Although  they confronted with the works of other authors whose diets did not contain any drug supplementation (antibiotic or coccidiostatic), they do not mention this important fact. Below is a list of all these points, which will  need to be revised in the study taking this fact into account.

• Please explain more clearly the aim and rationale of this study in the introductory section, considering the treatment of animals with antimicrobial agents?

• Add a critique of your study.

• Does this mixture of antimicrobial substances affect the intestinal microbiota? In the Discussion section, the author should explain the possibility of involvement (or non-inclusion) of the individual components of the antimicrobial substance mixture in the reduction of intestinal bacteria and their influence on the resulting intestinal fermentation.

• If additional experiments are to be performed in the future work, then the morphometry and functionality of the intestinal mucosa and the composition of the cecal microbiota should be monitored for a more objective monitoring of animal growth performance and food digestibility, as it is evaluated in other studies with a similar focus.